# Position: We Need AI Efficiency Incentives for Accessibility and Sustainability

**Marco Bornstein** [1]   **Amrit Singh Bedi** [2]

## Abstract

The race for artificial intelligence (AI) dominance often prioritizes scale over efficiency. Hyperscaling is the common industry approach: larger models, more data, and as many computational resources as possible. Using more resources is a simpler path to improved AI performance. Thus, efficiency has been de-emphasized. Consequently, the need for costly computational resources has marginalized academics and smaller companies. Simultaneously, increased energy expenditure, due to growing AI use, has led to mounting environmental costs. In response to accessibility and sustainability concerns, this position paper argues for research into, and implementation of, market-based methods that incentivize AI efficiency. We believe that incentivizing efficient operations and approaches will reduce emissions while opening new opportunities for academics and smaller companies. As a call to action, we propose a cap-and-trade system for AI. Our system provably reduces computations for AI deployment, thereby lowering emissions and monetizing efficiency to the benefit of academics and smaller companies.

## 1. Introduction

The advent of ChatGPT in November 2022 popularized large language models (LLMs) and kicked off a generative artificial intelligence (AI) craze. With the hype of generative AI raging at an all-time high, companies have heavily invested in LLM development and deployment in the hopes of (i) revenue generation, (ii) workplace automation and cost reduction, or (iii) becoming an industry leader within AI to capture market share. Simply labeling one's company as AI or AI-adjacent may spur massive private, consumer, or institutional investment even if the company does not develop AI or is generally unprofitable. This has led to a new, modern-day arms race: graphics processing units (GPUs).

If AI models are the car, and data is the fuel, then GPUs are the engines that make everything run. A large car (model) will not drive (train or infer) quickly if its engine (GPU compute power) is too small, regardless of how much fuel (data) is available. Today's LLMs are billions or trillions of parameters in size, but they require orders of magnitude more computations to both train and produce outputs (inference). The number of computations, denoted as floating point operations (FLOPs), required to train state-of-the-art LLMs is rapidly approaching a ronnaFLOP ($1e27$ FLOPs) (Epoch AI, 2025). Simultaneously, inference may be even more expensive. An average query with a combined prompt and response of 750 tokens (Zhao et al., 2024), though likely larger with reasoning models, uses $\approx 8e14$ FLOPs for an LLM utilizing half-a-trillion parameters. Therefore, a company like OpenAI, processing approximately 2.5 billion daily prompts (Silberling, 2025), conservatively uses around one ronnaFLOP *in inference per year alone*.

One top-of-the-line NVIDIA H100 GPU, valued at $30,000, has a maximum rating of 133.8 teraFLOPs per second ($1.3e14$/s) for 16-bit precision (NVIDIA, 2022). Even if the H100 could attain half of its maximum rating (Brown, 2025), it would take roughly 3 seconds per query on its own. With OpenAI receiving roughly 40,000 queries per second, at least 120,000 H100s are needed to ensure that queries are answered at the same rate they are received. The resulting cost would be 3.6 billion dollars, in GPU purchases, for inference alone. This does not even consider the energy nor infrastructure costs of AI deployment at such a large scale.

**Impeded Accessibility.** The sequestering of large-scale GPU power into the hands of a few companies impedes accessibility and threatens innovation. Without access to large-scale computational infrastructure, academics and smaller companies are priced out from competing on a level playing field. While academia still publishes a large amount of highly-cited research (Maslej et al., 2025), state-of-the-art AI models and performance almost exclusively stem from industry-leaders and their research (Eastwood, 2023; Maslej et al., 2025). Even as academics or smaller companies produce high-quality research, they face major headwinds. First, once new research is published, industry-leading companies simply adopt and implement it at larger scales. Second, even if academics or start-up entrepreneurs do not publish their research in hopes of developing a new product,

[1]Independent Researcher, Baltimore, MD, USA [2]University of Central Florida, Orlando, FL, USA. Correspondence to: Marco Bornstein <marcobornsteinresearch@gmail.com>.

*Proceedings of the 43rd International Conference on Machine Learning*, Seoul, South Korea. PMLR 306, 2026. Copyright 2026 by the author(s).

it is improbable to take such an innovation to market. Such a task would require billions of dollars to implement at scale. Third, access to computational resources and the allure of large pay packages (Isaac et al., 2025) has sparked the reality of an academic "brain drain"; nearly 70% of AI PhD graduates in 2023 took jobs in industry (Eastwood, 2023).

**Eroding Sustainability.** Beyond monetary costs, growing numbers of FLOPs come at an environmental risk. Each FLOP necessitates power and cooling costs. As a result, training and inference of LLMs has spurred a massive increase in electricity usage. Since the majority of electricity is not generated in a renewable manner, LLM usage is environmentally detrimental. The amount of watt-hours to answer an LLM query, like ChatGPT or Gemini, is estimated to be around 0.24-0.34 watt-hours (Altman, 2025; Elsworth et al., 2025). When processing 2.5 billion queries per day, OpenAI approximately expends 850,000 kWh per day. With an estimated 0.81 pounds of CO2 emissions per kWh in the United States (U.S. Energy Information Administration, 2024), OpenAI pollutes around 125,000 tons of CO2 emissions per year. For reference, the United States Environmental Protection Agency (EPA) requires facilities emitting over 25,000 tons of CO2 annually to report under its Greenhouse Gas Reporting Program (U.S. National Archives, 2025). OpenAI's inference emissions alone exceed this threshold by roughly five-fold annually. With AI usage increasing, its carbon footprint will continue to grow.

**In this position paper, we argue for the realignment of AI development and deployment away from its current hyper-scaling trajectory towards a smarter, more efficient approach. We argue that a market-based method is needed to properly incentivize companies to leverage more efficient operations and approaches for AI deployment. Such a method would allow academics and smaller companies to profit off their efficiency and reduce emissions. As a call to action, we propose our own market-based method, inspired by emission trading systems, to take a first step towards incentivizing AI efficiency.**

## 2. Related Work

**Market-Based Methods for AI Realignment.** The recent work of Tomei et al. (2025) reviews, and provides case studies for, market-governance mechanisms for responsible AI development. Like our paper, Tomei et al. (2025) argues for researchers to investigate and implement market-based approaches to AI governance. Closely related, Casper et al. (2025) argues for the enactment of "evidence-seeking" policies that will generate evidence of AI risks that can be used for long-term AI governance. Unlike Tomei et al. (2025) and Casper et al. (2025), we argue that market-based methods are needed to improve AI efficiency such that issues of accessibility and sustainability are alleviated. Furthermore, we

overview potential market-based methods for AI efficiency and propose our own market-based method to incentivize AI efficiency for improved accessibility and sustainability.

Similar methods have also been proposed for Artificial General Intelligence (AGI) safety. In Tomašev et al. (2025), market mechanisms underlie agent-to-agent transactions in order to mitigate safety risks of distributional AGI. Our paper focuses on solving issues of accessibility and sustainability within current AI development. Furthermore, as detailed in Section 3, solving such issues has positive effects on AGI safety. In Lior (2021), insurance, as a regulatory mechanism, is floated to mitigate the risks of AI damages and provide compensation for when damages occur. Insurance may be effective at reducing and preventing user harm, but it does not incentivize companies to develop and deploy more efficient AI models. Finally, Ball (2025) proposes a framework where leading AI companies can opt into, and thus must comply with, certifications regarding AI safety or security provided by private bodies (overseen by the government). In exchange, companies receive tort-liability protection from misuse of their AI products. Again, like Lior (2021), Ball (2025) proposes a framework centered on consumer use and not AI development or deployment.

**Market-Based Methods for Sustainability.** The foundational work of Porter & Linde (1995); Porter (1996) posits that environmental regulation, including market incentives, "reinforce resource productivity and also create incentives for ongoing innovation". The Porter hypothesis thereby argues that environmental market-based incentives spur companies to be more efficient and competitive (Ambec & Barla, 2002). The work of Stavins (2003) provides an overview of the effectiveness of differing market-based environmental incentives; such incentives have been successful in reducing pollution at low cost. Specifically, the success of tradable permit systems at reducing air pollution is detailed.

Investigating the impact of AI on the environment is critical to understand the scope of the problem. Simply put, one cannot incentivize what one cannot measure. Notably, Google and Mistral AI have released papers (Patterson et al., 2021; Elsworth et al., 2025; Mistral AI, 2025) quantifying the emission levels and water consumption of their AI. In Hebous & Vernon-Lin (2024), tax policy is proposed to curb emissions and air pollution from AI. A $0.052 tax per kWh would be levied on data centers to reduce electricity usage. While effective, a broad tax would unfairly penalize data centers relying on clean energy while also incentivizing companies to relocate to avoid such taxes. The United Nations' recent report (UNEP Copenhagen Climate Centre, 2025) advocates for usage- and outcome-based pricing adoption to incentivize efficiency. While effective at incentivizing efficiency at the user's end, it does not incentivize companies to develop and deploy more efficient AI models.

## 3. AI Without Market-Based Incentives

In the absence of market-based incentives, AI development has focused on growth at the expense of efficiency. Each year, companies are using larger models, more data, and as many GPUs as possible. We argue that growth and efficiency are not mutually exclusive. In fact, as with DeepSeek (Liu et al., 2024; Guo et al., 2025), we believe that proper efficiency incentives can spur growth-by-efficiency in AI.

**DeepSeek: A Case Study in Incentivized AI Efficiency.** We want to begin by noting that there exists incentive, albeit indirect, for companies to develop more-efficient AI: reducing energy costs. Google, for example, has reduced emissions per prompt by over 30x (Elsworth et al., 2025). While Google's pledge to reach net-zero emissions by 2030 (Clancy, 2025), a form of realignment itself, may play a role, driving down electricity and water costs undoubtedly boosts Google's bottom line. However, such incentives are indirect and insufficient. Energy costs are still rising year-over-year. In fact, the power required to train frontier AI models has grown by 2.4x per year since 2020 (Epoch AI, 2023).

In recent years, the United States has issued increasingly stringent export rules on computing chips (Bureau of Industry and Security, 2022; 2023; 2024) with the goal of impeding China's development of AI (Webster & Shen, 2025). The result has arguably been a strong, market-based incentive on efficiency for AI developers in China. Without access to the best compute infrastructure, Chinese developers have been incentivized to innovate efficient AI approaches to bridge the gap (Shivakumar et al., 2025; MERICS, 2025). Consequently, one such developer, DeepSeek, leveraged sparsity (Mixture-of-Experts) and compression (Multi-head Latent Attention) advancements to achieve performance on par with frontier AI models at a fraction of the training and inference costs (Dai et al., 2024; Guo et al., 2025). Such efficiency innovation, including recent advances in sparse attention (Liu et al., 2025), has been essential at driving down AI training and inference costs.

Even with a stiff, market-based impedance on compute power, DeepSeek showed that efficiency can power growth. We believe that small, and much less severe, market-based incentives can steer AI companies towards a growth-by-efficiency approach. Not only would this reduce underlying energy costs, which in turn bolsters balance sheets and reduces emissions, but it would also allow academics and smaller companies to remain competitive without the powerful compute resources of larger rivals.

**Rapid Emission Increases.** While we noted that carbon emissions per prompt have rapidly decreased, at least for Google (Elsworth et al., 2025), emissions are still rising year-over-year due to increases in consumer use of AI (Epoch AI, 2023; de Vries-Gao, 2025). In fact, increasing AI emissions have already forced Google to scrap its longstanding carbon-neutral policy (Rathi & Hirji, 2024).

The major driver of AI-related emissions are data centers, which power AI training and inference. Over the remainder of the decade, global electricity consumption by data centers is projected to grow 15% annually (International Energy Agency, 2025). The result will be more than doubling current data center electricity consumption, to nearly 1,000 TWh, by 2030. Simultaneously, water consumption in data centers is projected to also more than double, to 1,200 billion liters, by 2030. Overall, in the United States, data center emissions are projected to grow by 70% in the next decade, accounting for 3.3-4.5% of national combustion $CO_2$ emissions (International Energy Agency, 2025).

While manufacturing and data center construction emissions account for around 20-30% of lifetime AI hardware-related emissions (Schneider et al., 2025), the majority stem from operational emissions. Thus, as the demand for AI continues to expand, mitigating the main driver of AI-related emissions, training and inference, is critical for sustainability. Incentivizing AI efficiency is the most direct way to accomplish this goal.

**Imbalance of Power and Control.** The large cost of AI development and deployment, detailed in Section 1, has generated an economic barrier to entry for smaller AI companies. Not only would a state-of-the-art AI model require close to a ronnaFLOP ($1e27$ FLOPs) of compute power, wide-spread deployment of said AI model would necessitate a near ronnaFLOP annually too. Competing with the industry leaders, such as OpenAI, Google, or Anthropic, would thus require tens of billions of dollars in GPUs and a multi-billion dollar annual operating budget for inference costs alone (Isaac & Griffith, 2024; Mascarenhas & Palazzolo, 2025). Consequently, the market for state-of-the-art development and deployment of AI has become an oligopoly.

While Artificial General Intelligence (AGI) remains hotly debated, concerns surrounding its creation and ramifications are amplified in an oligopolistic setting. As model performance tends to improve as model and data size increase (Kaplan et al., 2020; Billa & Jing, 2025), it is most likely that the industry leaders, those with the most computational power, would be the first to reach AGI. One immediate concern in this scenario is trust. What happens if the goals of these few companies diverge from those of the general public? Will AGI be deployed in a harmful manner that reduces overall social welfare? Another immediate and pressing concern is safety. Is it possible for a government, or any other entity, to govern, regulate, or stop a company once it develops AGI? What checks on power are there if only a few have the ability to develop AGI? In summary, the current oligopolistic setting is at risk of providing a handful of companies with unmitigated power over the rest of society.

**A New Step Forward: Market-Based Methods.** Regulatory approaches have dominated the formulation and implementation of AI governance. Nevertheless, even regulatory frameworks are rare throughout the world. While regions, like Europe, have begun to advance measures to mitigate AI risk (European Union, 2024), there are still areas, like the United States, with limited to no AI oversight (GRC Report, 2025; The White House, 2025b). What is lacking within both current discourse and research into AI governance are market-based methods. Such methods have the promise of incentivizing efficient and responsible AI development and deployment while still enabling AI growth (Tomei et al., 2025). Now, we detail existing market-based methods that have been, or can be, applied to incentivize efficiency.

## 4. Market-Based Methods for Efficiency

### 4.1. Charge Systems

One basic market-based method for efficiency is to charge companies, or users, for inefficiency. The broader class of such methods is called a charge system.

#### 4.1.1. PIGOUVIAN TAXES

A Pigouvian tax is a tax on market transactions that create negative externalities. This is the simplest approach: penalize inefficiency through taxation. There is a breadth of research on Pigouvian taxes, and their efficacy, at reducing pollution (Porter & Linde, 1995; Rubio & Escriche, 2001; Metcalf, 2021). Furthermore, recent research proposes Pigouvian taxes for penalizing tech companies that look to capture human attention on social media (Belgroun et al., 2025). Within AI, Pigouvian tax scenarios include taxing electricity usage at data centers (Hebous & Vernon-Lin, 2024), carbon emissions resulting from AI development and deployment, or quantity of FLOPs used by each company.

#### 4.1.2. USER FEES

Similar to Pigouvian taxes, user fees can be enacted to incentivize efficiency. User fees are already implemented to improve health outcomes, such as increased sugar or alcohol sales taxes (Blecher, 2015), as well as environmental causes such as airline traffic taxes to finance noise pollution abatement (Porter & Linde, 1995). Interestingly, user fees have already been floated for AI use. In Korinek & Lockwood (2025), both general consumption taxes and a "token tax" are proposed to generate revenue and incentivize efficiency. The "token tax", a tax on AI-generated tokens sold to users, is a promising idea, one that penalizes wasteful AI use.

#### 4.1.3. CREDITS AND SUBSIDIES

Credits and subsidies, including tax breaks, continue to be used to reward energy efficiency (Panayotou, 2013; Allcott et al., 2015; European Commission, 2024b). Commonly, governments provide tax credits for usage of renewable energy sources (Espa & Rolland, 2015), purchases of electric vehicles, or energy-efficient homes (Internal Revenue Service, 2024). This too can be extrapolated to AI. Governments can reward companies, via credits and subsidies, that use clean energy or minimize FLOPs for AI development and deployment.

#### 4.1.4. DEPOSIT REFUND

A deposit-refund system is a combination of the previous two charge systems. A tax is levied on consumption of a product and a credit or rebate is returned when the product is properly disposed or replaced (Walls, 2011). The most common instance of a deposit-refund system is bottle recycling programs, where money is refunded after bottles are returned. Such programs are effective. In the United States, specifically Michigan, the return rate of bottles was 95% after one year of the program (Porter, 1983). Implementations of deposit-refund systems for AI could include companies paying a governing body an "environmental deposit" proportional to the energy consumption prior to AI training. Then, the deposit is partially refunded if the model is shown either to have been trained more efficiently or to perform efficient inference. Beyond efficiency, deposit-refund systems could be used for responsible AI. AI companies would pay a deposit, or potentially receive a loan for model development, that is returned with interest after a set period of time if the AI model has not caused any safety issues.

### 4.2. Tradable Permits Systems

Tradable permit systems, a relatively new approach, set a compliance level or cap on pollution in a geographic area. Companies in the area that pollute above a certain level are mandated to participate in the system. These companies can only pollute according to the compliance level or the quantity of permits that they have. There are two such types of systems: cap-and-trade and credit programs.

#### 4.2.1. CAP-AND-TRADE

Cap-and-trade is a system where companies receive emission allowances (the ability to pollute a certain quantity) annually from a governing body and are harshly penalized if they emit more than allowed. As such, emissions are capped by the total allowances provided by the governing body. Companies are free to trade their allowances with one another, but new allowances are never able to be generated.

**Allowance Distribution.** Many of the leading cap-and-trade systems around the world, such as in the EU (European Commission, 2024a), China (International Monetary Fund, Asia and Pacific Dept., 2024), California (California Air Resources Board, 2024), and South Korea (International

Carbon Action Partnership, 2025), distribute allowances for free. The alternative involves selling allowances, usually via auction, to companies. The rationale behind free allocation within these emissions trading systems (ETS) is financial. Free allocation aims to limit the defection of companies to areas with more lax regulations in order to reduce financial constraints. This phenomenon is called carbon leakage. While free, allowances are typically distributed proportionally among companies. The two common free-allowance distribution methods are grandfathering and benchmarking.

**Grandfathering.** In this simple approach, each company $i$ is allocated allowances based on its historical emission levels or from a baseline period $H_i$. Over time, the number of allowances $A_i$ is reduced by a scaling factor $\gamma \in (0, 1)$,

$$A_i = \gamma \times H_i. \tag{1}$$

While grandfathering is effective at curbing emissions, it is neither as effective at incentivizing investment in energy-efficient technology (Yang et al., 2020) nor the preferred option for policy-makers (Wang et al., 2022).

**Benchmarking.** The most common method of free-allowance distribution in existing cap-and-trade systems is benchmarking (Groenenberg & Blok, 2002; Zipperer et al., 2017). In benchmarking systems, a governing body allows each company $i$ to use its current level of output $O_i$ irrespective of other companies. For example, in ETS, the output could be tons of steel produced for a steel company. Uncapped output levels reduce the risk of carbon leakage.

What a governing body does dictate, however, is the *benchmark B* for how efficiently the output should be produced (Groenenberg & Blok, 2002). In ETS, the benchmark could be the amount of $CO_2$ released per ton of steel produced. Existing benchmarks use 90% of the average company efficiency (California Air Resources Board, 2010) or the top 10% in each product area (Lilico & Drury, 2023). Using such a benchmark rewards efficient companies and incentivizes companies to become more efficient.

Finally, an assistance factor $C_i$ is set for each company $i$. This assistance factor can simply be set uniformly as 1, or it can be smaller or larger for each company $i$. Assistance factors larger than 1 can be implemented to reduce carbon leakage or to reward clean energy generation (*e.g.,* a company uses hydropower). An assistance factor smaller than 1 could be used to penalize companies for unclean energy generation (*e.g.,* fossil fuels), previous-year cap-and-trade violations, or monopolistic actions. In total, allowances $A_i$ for each company $i$ are distributed as follows,

$$A_i = O_i \times B \times C_i. \tag{2}$$

**Allowance Markets.** After the initial allocation of allowances by a governing body, *i.e.,* the primary market, a secondary market for allowance allocation develops between companies. Within this market, companies are allowed to buy and sell allowances from one another. Companies that receive more allowances than they need, *e.g.,* ones that are more efficient than the 10% benchmark, are able to sell their allowances for cash. Conversely, companies that are short on allowances for their projected annual output, *e.g.,* inefficient companies, can buy allowances on the secondary market in order to avoid breaching their allowance cap. These markets are generally centralized exchanges, *e.g.,* the European Energy Exchange (European Energy Exchange, 2024) or the Korea Exchange (Ji-won, 2025). In summary, allowances gain monetary value within the cap-and-trade system due to the secondary markets that arise.

**Allowance Banking.** Not only are companies able to buy and sell allowances, they are also able to save unused allowances for future years (Practical Law, 2024). If future output exceeds the allowances received that year, a company can use its banked allowances to cover the difference.

### 4.2.2. CREDIT PROGRAMS

Credit programs, the lesser-known tradable system, work by establishing a baseline emission level for all companies. When companies emit beneath the baseline, they receive credits for the difference (Porter & Linde, 1995). Companies that exceed the baseline must purchase credits from other companies in a credit market. Unlike cap-and-trade, credit programs do not cap total emissions, and theoretically there can be infinite credits as a result. If new companies enter the market, they can all receive credits if they emit below the baseline level. Thus, for growing economies, total emissions can still rise whereas they are capped, and decreasing annually, in cap-and-trade.

## 5. Call to Action: Cap-and-Trade for AI

One possible solution to incentivize AI efficiency is through a cap-and-trade-style framework. Akin to current pollution regulatory programs, a cap-and-trade framework for AI could gradually dissuade over-utilization of FLOPs while maintaining a friendly climate towards AI development and deployment (Schmalensee & Stavins, 2017). Specifically, a governing body distributes an *allowance* of power units (*e.g.,* ten one-megawatt allowances) to all mandated companies (*i.e.,* companies using more than a prescribed number of FLOPs) that they can use annually on FLOPs for AI model deployment. The allowances are freely distributed by a governing body across the mandated companies (distribution is detailed in Section 5.1). In current pollution cap-and-trade systems, such allowances are named Carbon Allowances. Here, we coin the name of allowances within our AI cap-and-trade system as *AI Allowances*. Companies are able to save leftover AI Allowances for future use or sell them to

other companies in exchange for money. Any company that uses more allowances than allowed is harshly penalized and taxed for its violation.

**Cap-and-Trade for AI Inference.** We fear that a blanket cap on total FLOP usage for AI development would be harsh, unrealistic, and detrimental to state-of-the-art research. Instead, we propose an annual power cap for FLOP usage based solely on AI deployment, or inference, for consumer use. A cap on inference, and not training, still eases the consequences of unchecked FLOPs. Inference is the major revenue generator for companies such as OpenAI; consumers pay API or general access fees in order to obtain state-of-the-art LLM responses (Adebayo, 2025). As detailed further in Section 5.2, up-and-coming companies can leverage their AI Allowances as a new stream of revenue. The result is increased competition within the AI sector. Furthermore, research has shown that inference plays an outsized role in AI-related emissions (Schmidt et al., 2021; De Vries, 2023; Jegham et al., 2025).

Regulating AI through FLOPs already has precedent. Legislation in the European Union (EU) (European Union, 2024) and United States (Biden, 2023) has already incorporated FLOP counts for monitoring AI safety and risk. Below, we detail a realistic cap-and-trade framework, similar to previous frameworks for emissions (Environmental Defense Fund, 2018; U.S. Environmental Protection Agency, 2016), that is tailored for AI.

### 5.1. AI Allowance Distribution

We surmise that an issue similar to carbon leakage (Section 4.2.1) would arise if free allocation of allowances is not implemented within the AI sector. AI companies would move to other locations in order to avoid the financial costs of purchasing AI Allowances. We coin this phenomenon *AI leakage*. To avoid AI leakage, our cap-and-trade system freely allocates allowances to mandated companies.

**Fair Division via Benchmarking.** While allowances are free for all mandated companies, the number of allowances distributed to each company need not be equal. It would be unfair for a company like OpenAI, with hundreds of millions of existing users, to share the same number of AI Allowances as a newer, smaller company such as Harvey AI. Distributing a uniform number of allowances would likely cause companies to cap or cut back the number of users that they can service. Thus, uniform distribution artificially induces unnecessary economic harm, via revenue suppression, to leading AI companies. As detailed in Section 4.2.1, a benchmarking strategy can remedy this issue and reduce the risk of AI leakage. Instead of capping output levels, the output level in an AI cap-and-trade system would be the number of FLOPs each company uses (*e.g.,* the EU uses a two-year rolling average with 15% adjustments (European

Commission, 2024a)). In the case of AI, the benchmark can be the amount of watts needed per FLOP. As companies are incentivized to become more efficient, the benchmark will naturally decrease year-over-year, thereby reducing the total number of FLOPs used.

**AI Allowance Allocation.** Each company $i$ is provided a certain number of AI Allowances that it can use for FLOPs during AI inference, determined via Equation (2). Given a company $i$'s own FLOP-efficiency $E_i$ (*i.e.,* watts-per-FLOP), the number of FLOPs it can use $F_i$ given their AI Allowances is effectively $F_i = A_i/E_i$. In the next section, we detail the courses of action for companies that have run out of AI Allowances or have a surplus of them.

### 5.2. Importance of AI Allowance Markets

Overall, the cap-and-trade system incentivizes efficiency. Efficient companies are allocated more allowances than they need, thereby generating a new revenue stream: selling surplus allowances to inefficient companies. This will inevitably lead to increased competition within the AI sector. Smaller, yet more efficient, AI companies may generate revenue, and stay afloat, by selling their surplus allowances to their inefficient peers. This new revenue stream is critical for startups and other new companies, allowing them to invest more in compute infrastructure or weather the financial storm (*i.e.,* burn rate) as they break into the AI sector. Furthermore, with a newfound market-based alignment towards efficiency, the environmental costs detailed in Section 1 will be driven down.

---

**Framework: AI Cap-and-Trade Governance**

1: **Initialize:** Governing body determines the benchmark $B$
2: **Set Assistance:** Determine assistance factor $C_i$ for each company $i$
3: **Distribute:** $A_i \leftarrow$ Allocations via Eqn. (2)
4: **Market Operations (Yearly Cycle):**
5: **while** Compliance Year is Active **do**
6:    Companies may **Buy** or **Sell** AI Allowances
7:    Surplus AI Allowances may be **Banked** for future periods
8: **end while**

---

### 5.3. AI Cap-and-Trade Analysis

**The FLOP-Performance Relationship.** Model performance generally improves as model and data size increase (Kaplan et al., 2020; Billa & Jing, 2025). Using larger models and more data requires more FLOPs during training and inference. Thus, generally speaking, using more FLOPs will generate better performance. Letting the number of FLOPs used be denoted as $x$, we relate FLOPs to performance (*i.e.,*

minimizing loss) as $1/x^k$, where $k > 0$ is a parameter. The value of $k$ can be set larger if fewer FLOPs are required to reach a given loss level. Intuitively, this formulation means that using more FLOPs leads to a smaller loss.

**Characterizing Cost-per-FLOP.** While FLOPs may boost performance, they also incur a marginal cost. Performing FLOPs incurs electricity, cooling, and other infrastructural costs that are constant over time (Borenstein, 2005; Schittekatte et al., 2024). As a result, cost-per-FLOP can be modeled as a linear term. Letting $a > 0$ be the cost-per-FLOP constant, $ax$ is the cost of using $x$ FLOPs.

**Modeling Utility.** In a game-theory setting, rational agents (*e.g.,* tech companies) seek to maximize their utility $u$. In doing so, each company balances the trade-off between using more FLOPs $x$ to reduce loss and incurring more FLOP-related costs.

$$\max_{x \in \mathbb{R}_{\geq 0}} u(x) = -x^{-k} - ax. \quad (3)$$

---

**Theorem 1: FLOP Equilibrium (No Governance)**

Rational agents following the utility function in Equation (3) will utilize $x^*$ FLOPs annually for inference, where $x^*$ is defined as:

$$x^* = \arg\max u(x) = (\frac{k}{a})^{\frac{1}{k+1}}. \quad (4)$$

---

*Proof.* Solving $\nabla u(x) = 0$ results in $x^* = (\frac{k}{a})^{\frac{1}{k+1}}$. $\quad \square$

**Cap-and-Trade Utility.** An AI cap-and-trade framework introduces a second variable $y$ which denotes the number of FLOPs bought or sold. When $y > 0$, FLOPs have been sold while $y < 0$ denotes that FLOPs have been purchased. Since the FLOPs are bought or sold at a set price, the cost of buying or selling FLOPs is denoted by the linear term $by$, with $b > 0$. Furthermore, the maximum number of allowable FLOPs imposed on company $i$ is denoted as $F_i$. The value $F_i$ is determined in practice as the AI Allowances for company $i$ divided by its FLOP-efficiency $A_i/E_i$ (detailed in Section 5.1). Since utility is analyzed for a single year, allowance banking is not considered. The introduction of new parameters and constraints leads to the formulation of a constrained, concave non-linear optimization problem.

$$\max \ u(x, y) = -x^{-k} - ax + by \quad (5)$$
$$\text{subject to:} \quad x + y \leq F_i$$
$$x \geq 0$$

---

**Theorem 2: FLOP Equilibrium (Cap-and-Trade)**

Rational agents with the utility function in Equation (5) use $x^*$ FLOPs and buy/sell $y^*$ FLOPs annually for inference, where $x^*, y^*$ are defined as:

$$x^* = (\frac{k}{a+b})^{\frac{1}{k+1}}, \ y^* = F_i - (\frac{k}{a+b})^{\frac{1}{k+1}} \quad (6)$$

---

Due to space, we leave Theorem 2's proof in Appendix A.

---

**Remark 1: FLOP Reduction via Cap-and-Trade**

Rational companies following a cap-and-trade system (Theorem 2) provably use fewer FLOPs $x^*$ than under no cap-and-trade system (Theorem 1).

---

In Figures 1 and 2, we plot the theoretical equilibrium for a company taking part in our cap-and-trade framework across varying costs. The results in Figure 1 bolster Remark 1; FLOP usage is shown to indeed decrease under our cap-and-trade framework. Furthermore, Figure 2 showcases that our cap-and-trade framework may *improve* utility for participating companies.

## 6. Alternative Views

Market realignment of AI development and deployment must be done carefully. For many countries, AI development and deployment is critical for economic growth and national security. Governmental AI oversight runs the risk of tamping down on regional or even global AI dominance. Therefore, a realistic framework for AI realignment must thread the needle between mitigating inefficient AI consequences and preserving national AI interests and dominance.

**Governmental Oversight Harms Growth & Innovation.** There is longstanding debate over the extent, and subsequent effects, of governmental oversight into free markets. One common argument is that economic growth is hampered by governmental action within free markets (Broughel & Hahn, 2022; Coffey et al., 2020; Dawson & Seater, 2013; McLaughlin & Wong, 2025). It is estimated in Coffey et al. (2020) that regulatory restrictions have reduced economic growth by around 0.8% per year from 1980 to 2012. McLaughlin & Wong (2025) estimate that real Gross Domestic Product (GDP) drops by 0.37% with a 10% increase in state-wide regulatory restrictions within the United States. In summary, governmental oversight of AI could lead to slower economic growth for the sector.

Stifling innovation is a major concern with governmental oversight. Long-term economic growth is driven, in large part, by innovation (Solow, 1957; Romer, 1990; Gürler,

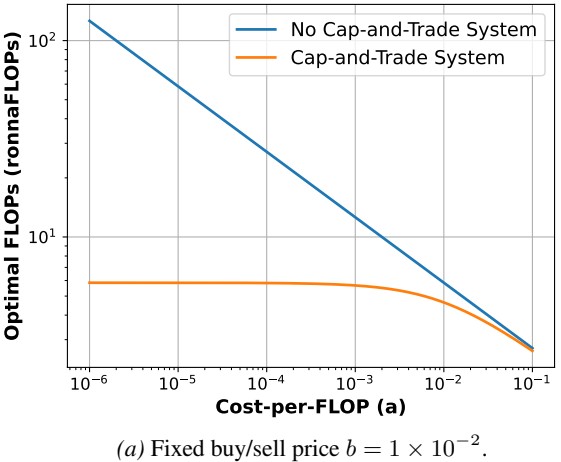
(a) Fixed buy/sell price $b = 1 \times 10^{-2}$.

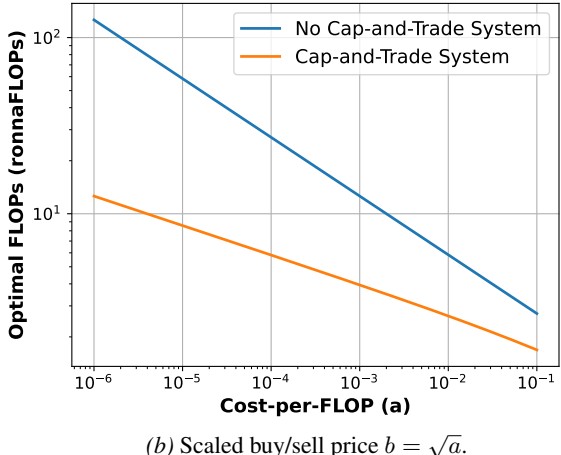
(b) Scaled buy/sell price $b = \sqrt{a}$.

*Figure 1.* **AI Cap-and-Trade System Reduces FLOP usage.** We verify that the number of FLOPs a company uses is always smaller in a cap-and-trade system. This is true for varying cost-per-FLOP values $a$ with fixed and scaled buy/sell price-per-FLOP $b$.

2022). In their review of cross-country studies, Broughel & Hahn (2022) posit that economic regulations impede competition and consequently innovation. Specific to AI, Bignami et al. (2025) found that the European Union's AI Act (European Union, 2024) poses large compliance burdens for companies. Companies in the EU spend up to $35,000 annually on compliance, excluding certification costs and other regulatory requirements (Bignami et al., 2025). While manageable for entrenched companies, these costs cause large barriers to entry for startups and smaller companies. When establishing market-based methods for AI, it is essential to construct them in such a manner that incentivizes innovation without being too costly for smaller companies.

**Unfettered AI is Essential for National Security.** As detailed in Section 3, many questions swirl regarding AGI and its potential repercussions on society. In a recent paper by the RAND Corporation, Mitre & Predd (2025) draw analogies between the development of AGI and nuclear weapons. While developing nuclear weapons was inevitable once nuclear fission was discovered, AGI remains an unknown. Mitre & Predd (2025) note that AGI presents national security risks, namely the possible emergence of wonder weapons, threatening and autonomous artificial entities, and development of dangerous weapons among non-experts. If large-scale compute is required to reach AGI, then growth-by-scale may be necessary to mitigate security risks.

Though now loosening (McGuire, 2026), the United States' strict chip-export policy to China, and other adversaries, is one example of a national policy geared towards security through AI dominance. In fact, global AI dominance is specifically mentioned by the United States as a national security imperative (The White House, 2025a). The White House (2025a) states that "to maintain global leadership in AI, America's private sector must be unencumbered by bureaucratic red tape". This encapsulates the alternative view: unimpeded AI development is necessary to protect national interests and security in the current AI arms race.

## 7. Conclusion

Currently, the AI sector is dominated by a growth-by-scaling approach, where companies build larger models and use more data and GPUs. This approach has impeded accessibility and eroded sustainability within AI. Namely, massive capital expenditure is required to compete in the AI sector, making research and deployment of state-of-the-art AI models inaccessible to academics and smaller companies. Additionally, growth-by-scaling encourages greater computational and, subsequently, energy usage. As much of the world's energy stems from non-renewable sources, a large uptick in pollution has resulted from increased AI usage.

Within this paper, we argue for market-based methods to realign AI development and deployment away from growth-by-scaling and more towards a growth-by-efficiency approach. We overview various existing methods, many of which have been applied in environmental settings, that have successfully incentivized efficient outcomes. Furthermore, as a call to action, we propose a novel AI cap-and-trade framework well-suited to incentivize AI efficiency. We derive an equilibrium for our framework, where companies use fewer computations for AI deployment. This would both generate new revenue streams for smaller companies breaking into the AI sector and reduce AI-related emissions. Finally, simulations of our equilibrium show that companies are indeed incentivized to use fewer computations while also improving company utility under certain scenarios.

## Acknowledgements

Bornstein wants to thank the late Dr. Howard Elman for his mentorship and career impact. Without him, Bornstein would not be nearly as motivated to continue academic research. We will deeply miss Dr. Elman, and may his memory be a blessing.

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

# Appendix

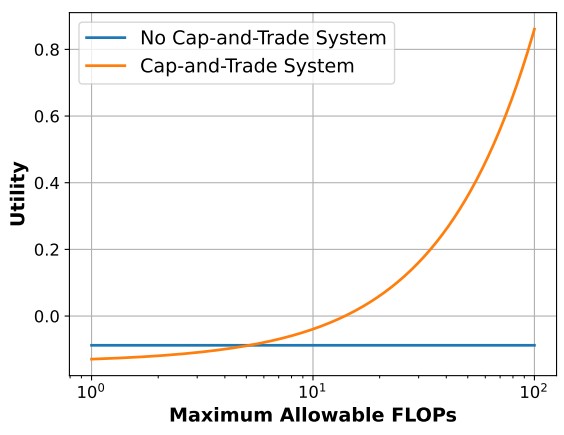

(a) Varying allowable FLOPs $F_i$, fixed cost $a = 10^{-2}$.

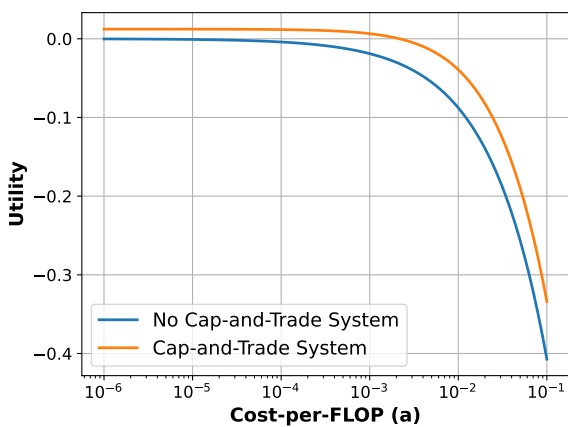

(b) Varying costs $a$, fixed allowable FLOPs $F_i = 10$.

*Figure 2.* **AI Cap-and-Trade Can Increase Utility.** When the maximum number of allowable FLOPs $F_i$ for company $i$ is large enough, our AI cap-and-trade framework *increases* utility compared to the current AI setting. This holds across various cost-per-FLOP values.

## A. Proof of Theorem 2

**Theorem 2: FLOP Equilibrium (Cap-and-Trade)**

Rational agents with the utility function in Equation (5) use $x^*$ FLOPs and buy/sell $y^*$ FLOPs annually for inference, where $x^*, y^*$ are defined as:

$$x^* = \left(\frac{k}{a+b}\right)^{\frac{1}{k+1}}, \ y^* = F_i - \left(\frac{k}{a+b}\right)^{\frac{1}{k+1}} \tag{7}$$

*Proof.* Since the problem is constrained with inequalities, we find an equilibrium using Karush-Kuhn-Tucker (KKT) conditions. Converting to a minimization problem (with a sign flip), we begin with our Lagrangian,

$$\mathcal{L}(x, y, \mu) = x^{-k} + ax - by + \mu_1(x + y - F_i) - \mu_2 x. \tag{8}$$

Solving for the first-order conditions yields,

$$\nabla \mathcal{L}(x, y, \mu) = \begin{bmatrix} -kx^{-(k+1)} + a + \mu_1 - \mu_2 \\ \mu_1 - b \end{bmatrix} = 0. \tag{9}$$

Solving each row, starting with the second row, yields,

$$\mu_1 = b, \quad x^* = \left(\frac{k}{a + b - \mu_2}\right)^{\frac{1}{k+1}}. \tag{10}$$

Due to complementary slackness, and since $\mu_1 = b > 0$, the first inequality must be tight,

$$x^* + y^* - F_i = 0 \longrightarrow y^* = F_i - x^*. \tag{11}$$

Looking at the objective function in Equation (5), the objective value is negative infinity if $x^* = 0$. Thus, it must be that $x^* > 0$. Due to complementary slackness, it is the case that $\mu_2 = 0$. Therefore we find that,

$$x^* = \left(\frac{k}{a+b}\right)^{\frac{1}{k+1}} \tag{12}$$

$\square$

