# OpenReview forum: "Position: We Need AI Efficiency Incentives for Accessibility and Sustainability"
_ICML.cc/2026/Position_Paper_Track — ICML 2026 Position Paper Track regular_

### Official Review · Reviewer_q1Xb · 2026-02-14

**Significance:** 3
**Argument Clarity:** 4
**Rating:** 5
**Confidence:** 4

**Questions:**

This is a well-written paper that proposes an interesting and important topic. Based on the alternative perspectives discussed in the paper, I would like to ask how the authors envision achieving an appropriate trade-off between the AI arms race (i.e., the rapid pace of AI development) and mitigating “growth-by-scaling.” In particular, how can such a balance be achieved without significantly hindering innovation, competitiveness, or practical deployment?

**Alternative Views Section:**

Yes

**Compliance With Llm Reviewing Policy A Conservative:**

Affirmed.

**Discussion Potential:**

4

**Paper Summary:**

This position paper argues that the current AI industry focus on "growth-by-scaling" (bigger models, more data, more GPUs) is economically exclusionary and environmentally unsustainable. The authors state that without intervention, high compute costs will continue to push academics and small companies out of the market.

To address this, the paper proposes a transition to "growth-by-efficiency" using market-based incentives. Specifically, the authors advocate for a Cap-and-Trade system applied to AI inference.

The authors support their proposal with empirical simulations, demonstrating that the framework effectively drives down computational usage. The results also suggest that under specific conditions, this system can actually enhance a company's overall utility.

**Position:**

Yes

**Position In Title:**

Yes

**Related Work:**

4

**Strengths And Weaknesses:**

Strengths:

- The paper is well written and easy to follow.
- The stated position is clearly articulated and represents an important topic worthy of discussion.
- The authors provide multiple pieces of evidence and supporting resources to substantiate the importance of their stated position.
- The authors propose a potential solution to mitigate AI “growth-by-scaling,” and they provide both theoretical and empirical validation of its effectiveness.


Weaknesses:

- The proposed solution relies on a relatively simple economic model. It assumes rational agents acting solely on FLOP costs and efficiency, which may lack "real market sense" by failing to capture the complex, multi-faceted drivers of corporate behavior in the AI industry.
- The proposal relies on establishing a fair benchmark (e.g., Watts-per-FLOP) to distribute allowances. However, implementing this in practice is extremely difficult given the vast heterogeneity of hardware (GPUs vs. TPUs) and diverse AI architectures.
- While the authors candidly state the alternative views—specifically that regulation might stifle innovation or disadvantage national security in an AI arms race—they do not provide concrete mechanisms to mitigate these risks.

**Support:**

4

---

> ### Author Rebuttal · Authors · 2026-03-30
>
> Dear Reviewer q1Xb,
>
> Thank you for your thorough and thoughtful review. Below we address all weaknesses and questions.
>
> ## Weaknesses
>
> ### Economic Model
>
> > **W1:** The proposed solution relies on a relatively simple economic model. It assumes rational agents acting solely on FLOP costs and efficiency, which may lack "real market sense" by failing to capture the complex, multi-faceted drivers of corporate behavior in the AI industry.
>
> **(Response to W1):** Thank you for pointing this out. Our theoretical model is a good starting point to establish the position we are trying to make in this work. For now, we include the necessities: agent payoff (model performance $1/x^k$) and cost ($ax$). Future research can expand into more complex, non-linear, and multi-variable analysis.
>
> ### Benchmarking Feasability
>
> > **W2:** The proposal relies on establishing a fair benchmark (e.g., Watts-per-FLOP) to distribute allowances. However, implementing this in practice is extremely difficult given the vast heterogeneity of hardware (GPUs vs. TPUs) and diverse AI architectures.
>
> **(Response to W2):** We first want to clarify that the benchmark $B$ is used to determine the number of AI Allowances each company receives, and does not mandate that each company use hardware that meets the benchmark. Simply put, companies with less efficient hardware that does not meet the benchmark will be allocated fewer AI Allowances (a penalty thereby incentivizing the procurement and use of more efficient hardware or inference methods).
>
> For example, the regulator can set the benchmark as the 90th percentile of Watts-per-FLOP for current hardware, and companies will be allocated a corresponding number of AI Allowances that they can use regardless of their hardware.
>
>
> ### Alternative Views (Also Question 1)
>
> > **W3:** While the authors candidly state the alternative views—specifically that regulation might stifle innovation or disadvantage national security in an AI arms race—they do not provide concrete mechanisms to mitigate these risks.
>
>
> > **Q1:** This is a well-written paper that proposes an interesting and important topic. Based on the alternative perspectives discussed in the paper, I would like to ask how the authors envision achieving an appropriate trade-off between the AI arms race (i.e., the rapid pace of AI development) and mitigating “growth-by-scaling.” In particular, how can such a balance be achieved without significantly hindering innovation, competitiveness, or practical deployment?
>
>
> **(Response to W3 & Q1):** This is an important and insightful question. **The answer lies within the "cap" in the cap-and-trade framework: benchmark and assistance factors must be carefully selected such that there is a balance between innovation/security and efficiency incentivization.**
>
> A regulator can effectively keep FLOPs "uncapped" by setting large values for the benchmark $B$ and assistance factor $C_i$ in Equation 2 (this results in excess AI Allowances for a company). Consequently, the status quo would remain: efficiency is not incentivized. Inversely, FLOPs can be strictly "capped" by selecting overaggressive benchmark and assistance factor values. Consequently, efficiency would be hyper-incentivized at the cost of innovation and national security.
>
> In practice, a policy and mechanistic approach can solve this issue. First, a regulatory commission or report would be necessary to guide the regulator towards the exact benchmark and assistance factor values to use such that innovation/security and efficiency incentivization are balanced. Second, a phased mechanism (common among emission trading systems) can be implemented. In the phased mechanism, the original "cap" distributes a generous amount of AI Allowances that impose minimal constraint. Then, the "cap" is slowly tightened based on observed market behavior and efficiency gains. We will add a discussion of phased implementation within the paper.

---

> > ### Author Rebuttal · Reviewer_q1Xb · 2026-04-05
> >
> > The authors have addressed most of my concerns in their rebuttal with appropriate clarifications. I will keep the positive score.

---

### Official Review · Reviewer_1XRm · 2026-02-19

**Significance:** 3
**Argument Clarity:** 3
**Rating:** 3
**Confidence:** 4

**Questions:**

The The FLOP-Performance Relationship described seems to decrease with the flops x. $1/x^k$ with $k>0$. What does this formula actually describe?

What is meant by loss and performance in this paragraph?

Why is the system necessary at all?
For a fixed cost per inference $a$, there might the equilibrium described in Theorem 1. But generally, $a$ is not fixed as it can be reduced by adopting the very measures we aim to incentivize: smaller models and more energy efficient hardware.

**Alternative Views Section:**

Yes

**Compliance With Llm Reviewing Policy A Conservative:**

Affirmed.

**Discussion Potential:**

4

**Final Justification:**

The general topic is interesting and causes a lot attention in the machine learning community which is a good thing for a position paper. Nonentheless, the hypotheses in this paper are built on a simplicistic view on where the money from AI training comes from.

**Paper Summary:**

The paper proposes certificate markets for AI companies in order to be allowed to spend computing power in inference. The authors argues that the enormous growth of inference costs for AI services leads to increasing threats to the worldwide energy consumption and the associated environmental impacts. A further argument is that small companies cannot enter the market due to the enormous investments in compute hardware in order to generate a state-of-the-art AI model and offering users to do inference.
To adress these problems, the authors propose to hand out compute certificates based on efficiency benchmarks for inference. Under the assumption that the service improvemt does scale sublinear with more compute and given that costs for inference rise linearly this leads to a sweet spot. Furthermore, the authors also add the constraint that the used FLOPs have to add up to allowances of the companies plus allowances bought from the market.

**Position:**

Yes

**Position In Title:**

Yes

**Related Work:**

4

**Strengths And Weaknesses:**

Strong points:
The paper adresses an important problem as the number of computing power required to train state of the art AI models is increasing and the switch from search engine technology to AI based assistents yields an enormous increase of energy consumption for these services. I also agree that creating additional incentives for energy efficient inference might be required to keep this problems at bay.
Also, the authors made a very good job in surveying and explaining existing incentive systems.

Weak points:
1. The FLOP-Performance Relationship
The used relationship between performance and computing power is unclear. The formula is decreasing with more FLOPs which contradicts the introductionary text. On a more general level, the authors seem to talk about benchmark performance and not about the monetary profit made with inference. As the other terms in their optimization problem seem to refer to monetary costs for running inference and buying certificates, the term should also be built on the income per user query.
The need for the proposed Cap-and-Trade system is strongly dependend on the assumption that performance corresponds to income and increases uncapped with additional computing power. The authors should discuss more clearly why this is the case.
Otherwise, it is likely that the only way to increase profit is to lower the cost per inference a, even without a Cap-and-Trade solution.

2. Unstable Business Models
Many inference-heavy AI products like ChatBots are currently provided on a non profitable basis. For example, for ChatGPT most user queries are from free users, most of the systems are financed by investor's expectation that the better product will replace inferior services at some. However, though users switch more and more from search engines to chatbots for their information needs, companies are still struggling with finding the right way to make the cash machine out of the service that investors are expecting.
One way to increase the profit is reducing the cost. Thus, once the business model is adult enough to sustain the service without injecting money, companies most likely will go for the cheapest model capable providing the user's needs instead anyway.

3. Implementation of the proposed method
For CO2 certificates, a general idea is that every country is part of this planet and if some over exploit the environment the damage is done to everyone. Thus, everyone should have a share. For AI, this is more sensible as who is deciding who is eligable to earn certificates. In other words, companies might be founded with the sole purpose to generate tradable certificates without actually offering a real AI services. Though regulation might be installed that avoid this problem, avoiding any loopholes is problematic.

4. Double Pricing
As the Flop certificates are coupled to energy prices and these again are already coupled to CO2 pricing, the system might imply that AI companies would have to pay twice for using energy which is unfair compared to other areas which have even larger energy consumption und CO2 emission (e.g. air travel).



Conclusion: Though the paper raises an interesting question on whether a system like this might be beneficial, some details of the proposed Cap-and-Trade system are not explained enough.

**Support:**

2

---

> ### Author Rebuttal · Authors · 2026-03-30
>
> Dear Reviewer 1XRm,
>
> Thank you for your thorough and thoughtful review.
>
> ## Position Paper Remark
>
> > Conclusion: Though the paper raises an interesting question on whether a system like this might be beneficial, some details of the proposed Cap-and-Trade system are not explained enough.
>
> We want to emphasize that the goal of our position paper is to indeed "raise an interesting question on", and argue for, the enactment of market-based method to incentivize efficient AI deployment.
>
> * The ICML guidelines state "position papers make an argument for a viewpoint or perspective about what *should* be done, in contrast to main track papers, which report on advances that have already been accomplished" and that position papers are " judged primarily on whether they present a compelling position that warrants greater exposure within the machine learning community *(regardless of whether a reviewer agrees with the position)*".
> * While we propose a cap-and-trade framework as a call to action, *this is taken as a first step* and should not need to meet the "accomplished advancement" requirement of main-track papers.
>
>
> ## Questions
>
> Below we address all of the questions raised in the review.
>
> ### FLOP-Performance Relationship
>
> > **Q1.1:** The The FLOP-Performance Relationship described seems to decrease with the flops $x$. $1/x^k$ with $k>0$. What does this formula actually describe?
>
> **(Q1.1 Response):** We denote $1/x^k$, $k>0$ as the relationship between the number of FLOPs $x$ and the loss of an AI model. In sum, model loss scales as a power-law with the number of FLOPs.
>
> * This relationship is backed by our cited work (Kaplan et al. 2020), which states LLM "loss scales as a power-law with model size, dataset size, and the amount of compute used for training".
> * Furthermore, Hoffmann et al. (2022) [C1] demonstrates that training loss decreases monotonically with respect to FLOPs (Figure 2).
> * While instances of models trained with fewer FLOPs outperforming those with more exist, the general relationship is demonstrated to be true in foundational papers (Kaplan, Hoffmann).
>
>
> > **Q1.2:** What is meant by loss and performance in this paragraph?
>
> **(Q1.2 Response):** Many possible definitions of performance for AI models exist, but in this section we define the performance of an AI model as its loss on a given task. The lower the loss, the better an AI model performs. Thus, as FLOPs increase, the loss decreases and model performance increases.
>
> ### Framework Necessity
>
> > **Q2:** Why is the system necessary at all? For a fixed cost per inference $a$, there might the equilibrium described in Theorem 1. But generally, $a$ is not fixed as it can be reduced by adopting the very measures we aim to incentivize: smaller models and more energy efficient hardware.
>
> **(Q2 Response):** As detailed in Theorem 1, the derived equilibrium denotes how many FLOPs are used by an agent *"annually"*. While $a$ would indeed decrease over time due to improved efficiency efforts, it is unlikely to decrease over the course of a year (since new hardware and deployment methods take multiple months if not years to procure and implement).
>
>
> ## Citations
>
> [C1] Hoffmann, Jordan, et al. "Training compute-optimal large language models." 2022.

---

### Official Review · Reviewer_ubWa · 2026-03-08

**Significance:** 3
**Argument Clarity:** 3
**Rating:** 4
**Confidence:** 4

**Questions:**

1. What/Who would the governing body be (L263)? Do you agree that such a system would work best it if would be implemented on a world-wide scale? If only one major economy would implement such a system, other economies could be better off without "restrictions". It has to be world-wide such that companies do not move to non-regulated places, right?
2. When *allowance* is distributed (L262-266), does that only target companies? Are universities excluded? OpenAI used to be a non-profit. What about those? What about research centers?
3. Are you familiar with the rebound effect? If we increase the efficiency of a good/service, it could lead to an increased use of it, and thus to increased environmental impact. What is your take on this?
4. Consider the analog of a car. Someone buys a car (hardware) and has to pay a periodic fee/tax for having it and fuel (electricity) for using it. Why should we put an additional price on distance (FLOPs) instead on fuel (electricity)? Wouldn't a regulation/an additional fee of energy-use on data centers me more meaningful?

**Alternative Views Section:**

Yes

**Compliance With Llm Reviewing Policy A Conservative:**

Affirmed.

**Discussion Potential:**

4

**Final Justification:**

The authors have addressed most of by concerns.

I still do not understand how their approach helps in terms of accessibility. Their answer did not address my concern. Note that accessibility is also one of their central concepts and is named in the title!

Nevertheless, I slightly raised my scores for discussion potential and rating.

**Paper Summary:**

SOTA AI is driven by scale: larger models, more compute, more data. Scale has brought us impressive results but also comes with problems. **This position paper identifies/focuses on two problems: accessibility and sustainability.** Only a handful of companies have the resources to compete in developing SOTA models which leads to an accessibility problem for smaller companies and academia. When it comes to sustainability, beyond large monetary costs for buying and operating hardware (energy), there are rising environmental costs (CO2 per kWh, water usage) for running data centers.

**The position is that the current trend of hyper-scaling should move to a smarter, more efficient approach.** The authors argue for a marked-based method which incentivizes companies to invest and develop more efficient AI models. Their call to action is their own market-based method which is inspired by an emission trading system with the goal of incentivizing AI efficiency.

**Position:**

Yes

**Position In Title:**

Yes

**Related Work:**

3

**Strengths And Weaknesses:**

### Strengths

- The paper considers a very timely topic. The considered problem is valid.
- Their motivation of accessibility and sustainability problems of AI is good.
- The paper is well written.
- Convincing case studies, e.g., DeepSeek (L119ff).


### Weaknesses

The authors argue for their marked-based method to incentivize AI efficiency for improved accessibility and sustainability (see, e.g., L89-91, L106-57, L155-157).
- **Who is targeted?** To me, it is unclear whether only companies are targeted or whether also universities and research institutes or even non-profit organizations are also targeted (assuming they exceed the prescribed number of FLOPs (L262-266)).
	- I do not think academia (universities) should have to deal with allowances, especially when there is a market system where allowances are sold and bought in a monetary way. This would be to the disadvantage to academia in general.
	- This makes a difference. If only for-profit companies are targeted, I see how smaller companies can benefit (L291-294) but how would academia benefit? This does not help in terms of accessibility.
- **Accessibility.** They argue that academics and smaller companies face problems competing with larger players (L42-44). I agree with that point, but in my opinion, their proposed solution targets only smaller companies and not academia.
	- Under their model, larger companies can but allowances from smaller companies which
		- creates an incentive for the large company to be more efficient and
		- creates a new revenue stream for smaller companies who sell their allowances (because of efficiency or not using them) which creates a new revenue system for them (L291-302).

	However, how does academia (or other non-profit players) benefit here when it comes to accessibility?
- **Sustainability.**
	- It is unclear what/who a governing body would be (L263) and whether it would be sufficient if such a marked-based system is implemented in a major economy or whether the system would have to function world-wide.
	- Assume a company buys infrastructure to train and run their AI model. This causes large hardware costs (L36). In addition, it causes energy and cooling (water) costs (L111-124). From a sustainability point of view, one could argue that the hardware already has a large **embodied emissions** footprint. Thus, it is meaningful/beneficial to use the hardware as long as possible and as good as possible. In other words, such a large compute system should not idle and consume resources without meaning; it should be used. If a governing body distributes *allowance*, e.g., ten one-Megawatt allowance or $x$ FLOPs (L262-266), we might see systems idling, e.g., when no budget is left to buy more allowances from the market. **This seems counterproductive.**
- **Are FLOP limitations to way to go?** I am not sure why the traded allowance should be FLOPS. Consider the analog of a car. Someone buys a car (hardware) and has to pay a periodic fee/tax for having it and fuel (electricity) for using it. Why should we put an additional price on distance (FLOPs) instead on fuel (electricity)? Wouldn't a regulation/an additional fee of energy-use on data centers me more meaningful?
- **Disagreements.**
	- The statements *"using more FLOPs will generate better performance"* (L326-328), *"we relate FLOPs to performance"* (L328-311), and *"using more FLOPs leads to a smaller loss"* (L332-334) are IMO not true in the generality in which they are presented. There is a cap and no infinite scaling.
	This assumption is central in their model.
	- *"Performing FLOPs incurs electricity, cooling, and other infrastructural costs that are constant over time"* (L337-340). Why is electricity constant in time? Energy mixes and costs change in time.
	- *"We validate our framework empirically, finding [...]"* (L435-438) sounds like they actually have empirical results but it seems their results are merely simulations under some assumptions. Thus, the sentence is misleading.
- **Call for Action.** The proposed mechanism does not seem to apply to/target academia. Furthermore, the call for action seems to be targeted at policy-makers, not academia. Industry does likely not want to be regulated. Thus, I do not see how the call for action affects the ML/AI community. What shall we as a community do with the position paper? This venue might be the wrong place for it.


### Comments
- L47-48: broken sentence?
- L76: use misuse?
- L326: AI AI
- L301: weather?

**Support:**

3

---

> ### Author Rebuttal · Authors · 2026-03-30
>
> Dear Reviewer ubWa,
>
> Thank you for your thorough and thoughtful review.
>
> ## Questions
>
> Below we address all of the questions raised in the review.
>
> ### Who is the Governing Body?
>
> > **Q1:** It has to be world-wide such that companies do not move to non-regulated places, right?
>
> **(Q1 Response): Regardless of where companies are located, they must satisfy the rules and regulations of the regions and nations in order to service customers in those areas.**
>
> * For example, the General Data Protection Regulation (GDPR) and EU AI Act apply to companies whose AI products or services are used by EU residents regardless of corporate headquarters location [C1].
> * In the case of Cap-and-Trade, California requires a " deliverer of electricity ... has a compliance obligation for every metric ton of CO2 emissions ... when such emissions are from a source in California or in a jurisdiction where a GHG emissions trading system has not been approved for linkage" [C2]. Thus, companies delivering electricity from an area not falling under a linked (recognized) cap-and-trade system, they must comply with California's.
>
>
> **World-wide governance is not required, and such a world-wide governing body would be unrealistic. Instead, like many Cap-and-Trade programs, governance at a regional, state-wide, or national level is most pragmatic**.
>
>
> ### Who is Targeted?
>
> > **Q2:** When allowance is distributed (L262-266), does that only target companies?
>
> **(Q2 Response): Since we propose a general framework within our call to action, policy specifics are left up to the eventual regulating body.**
>
> * The designation of entities that must comply is a policy decision that can vary region to region. Thus, it falls out of the scope of our position paper.
> * A simple approach would be that each entity (including academic institutions) applies for a compliance exemption from the regulating body.
> * In principle, if we ran a regulating body, we agree that academic institutions and many research centers and non-profits should generally be considered exempt. Often this point is moot, most of these entities do not use nearly the same level of FLOPs as frontier AI institutes.
>
> ### Rebound Effect
>
> > **Q3:** Are you familiar with the rebound effect... What is your take on this?
>
> **(Q3 Response): Benchmarking is an effective tool to mitigate the rebound effect**. The regulating body can annually decrease the benchmark $B$ in Equation 2 to offset rebound effects if they occur.
>
> ### Car Analogy
>
> > **Q4.1:** Consider the analog of a car... Why should we put an additional price on distance (FLOPs) instead on fuel (electricity)?
>
> **(Q4.1 Response): Using this analogy, the AI Allowances distributed by our framework act as the currency (credits) used at the gas station to purchase fuel for a car.** Car owners have an allocated amount of credits to spend on fuel (electricity) annually. If they run out, they can purchase credits from other car owners. AI Allowances are in units of energy ("a governing body distributes an allowance of power units" -- Line 263) and not FLOPs. Thus, our framework indeed regulates the energy *used* for FLOPs.
>
> > **Q4.2:** Wouldn't a regulation/an additional fee of energy-use on data centers be more meaningful?
>
> **(Q4.2 Response): An additional fee on energy-use would unfairly punish all companies, those both efficient and inefficient. Our proposed method incentivizes efficient behavior, yet does not add costs for smaller or more efficient companies.**
>
> ## Weaknesses
>
> Due to space limitations, we only address the call to action weakness.
>
> ### Call for Action
>
> > **W1:** The proposed mechanism does not seem to apply to/target academia... What shall we as a community do with the position paper?
>
> **(W1 Response): Our position paper argues for the enactment of market-based method to incentivize efficient AI deployment. While we propose a potential framework as a call to action, more research by the academic community is necessary to determine the "optimal" market-based method.**
>
> * Academics will play an important role in the continued design and analysis of potential market-based frameworks for incentivizing AI efficiency.
> * *Furthermore, industry is still part of the ML/AI community.* Even if industry is hesitant to be regulated, arguing for efficiency incentives in AI deployment is a relevant position within the ML/AI community.
>
>
> ## Citations
>
> [C1] Prystai, Rostyslav. "EU AI Act 2026 Updates: Compliance Requirements and Business Risks." Legal Nodes, 2026.
>
> [C2] "Emission Categories Used to Calculate Compliance Obligations." Cal. Code Regs., tit. 17, § 95852. Legal Information Institute, Cornell Law School.

---

> > ### Author Rebuttal · Reviewer_ubWa · 2026-04-01
> >
> > First of all, I thank the authors for their answers.
> >
> > Personally, I still have some problems with the line of arguments the authors use but this is rather to their benefit as it shows discussion potential.
> >
> > - Do I understand the authors correctly that they do not fear that companies could move to lesser regulated places in the world because if they want to provide their service in a certain region, than they have to follow the regulation of that region. Thus, *fleeing companies* are not really an issue?
> > - Regarding Q4.2: The environment does not care whether a company is efficient or not. It cares about pollution, waste, water usage, etc. If energy that is used for data centers had it's own price, it would surely affect all companies. But smaller companies would probably also use less compute. And if a company wants to save on energy costs, they have to become more efficient. Wouldn't such an approach fix the same problem while being easier to implement? If usage * efficiency = cost, then a company can use more while maintaining the cost by improving their efficiency.
> >
> > I invite the authors to use the additional space for addressing the remaining part of the weaknesses which were skipped due to a lack of space.
> >
> > - **Weakness 2:** I get the sustainability aspect but how does the proposed approach help in terms of accessibility?
> > - **Weaknesses 3, 5, 6:** What is the authors take on this?

---

> > > ### Author Response · Authors · 2026-04-02
> > >
> > > Dear Reviewer ubWa,
> > >
> > > Thank you for your quick reply. We appreciate your discussion, and reply to your comments below.
> > >
> > > ## Questions
> > >
> > > > **Q1:** Thus, fleeing companies are not really an issue?
> > >
> > > **(Response to Q1):** In our rebuttal we wanted to emphasize that regulations must be complied with even if a company flees (*i.e.,* AI leakage). As noted in our paper (Lines 311-312), one way to mitigate AI leakage is to freely allocate allowances. While fleeing (leakage) occurs, a recent study on emission trading systems (ETS) finds that it "does not dismiss the possibility that firms react to environmental policies locating activity elsewhere, but it suggests that this is a more remote possibility than previous evidence suggested" [C1]. Furthermore, in [C1], it is mentioned that leakage is in fact reduced when using free allowances.
> > >
> > > Another safeguard to reduce the likelihood of leakage also exists. As detailed in our rebuttal to Reviewer q1Xb, a *phased mechanism* can be implemented where the original "cap" distributes a generous amount of AI Allowances that impose minimal constraint. Over time, the "cap" is slowly reduced, easing the impact on companies and dissuading them from fleeing.
> > >
> > > > **Q2:** And if a company wants to save on energy costs, they have to become more efficient. Wouldn't such an approach fix the same problem while being easier to implement? If usage * efficiency = cost, then a company can use more while maintaining the cost by improving their efficiency.
> > >
> > > **(Q2 Response):** This is an alternative method, and we detail this exact approach in Section 4 (Pigouvian Tax). Our paper's position is that market-based methods are needed to incentivize AI efficiency for accessibility and sustainability. While our cap-and-trade framework serves as our Call to Action, the point of our position paper is to garner discussion into market-based methods such as the Pigouvian Tax.
> > >
> > > We do reiterate that as opposed to a Pigouvian Tax, our cap-and-trade framework can address the issue but in a fairer manner: the burden falls predominantly on less efficient companies. We agree that both methods would fix this same problem (*i.e.,* efficiency incentives).
> > >
> > > ## Weaknesses
> > >
> > > We note that our response to W6 in the rebuttal was incorrectly marked as W1. Issues of targeting (W1) were covered in our rebuttal as well as here. Thus, we only reply to W2, W3, and W5 below.
> > >
> > > > **W2:** I get the sustainability aspect but how does the proposed approach help in terms of accessibility?
> > >
> > > **(Response to W2):** Incentivizing efficiency will allow academia to have greater accessibility and impact towards how frontier AI models are built. Academic research will power the growth-by-efficiency approach, whereas it is slightly de-emphasized in the current growth-by-scaling setting. This will also stem part of the academic "brain drain" by reducing some of the pressure for researchers to move to industry in order to influence frontier model development. Finally, the proposed approach can lower barriers-to-entry for academic ideas to translate into startups or other real-world deployment applications.
> > >
> > >  > **W3:** ... we might see systems idling, e.g., when no budget is left to buy more allowances from the market. This seems counterproductive.
> > >
> > > **(Response to W3):** As touched on in our Q1 response, our cap-and-trade framework can *phase* the "cap" such that companies are slowly constrained over time. This will allow companies to adjust their hardware over a longer period while incentivizing them to purchase more efficient hardware in the future. Existing ETS have been able to effectively balance this risk while reducing negative externalities like pollution (Stavins 2003; Schmalensee & Stavins 2017).
> > >
> > > We note that the first bullet of W3 (*i.e.,* world-wide governance) is covered in our original rebuttal.
> > >
> > > > **W5.1:** "using more FLOPs will generate better performance" (L326-328)... are IMO not true in the generality in which they are presented.
> > >
> > > **(Response to W5.1):** Due to space, we refer you to "Deduced Formula" in our rebuttal for Reviewer afca.
> > >
> > > > **W5.2:** Why is electricity constant in time?
> > >
> > > **(Response to W5.2):** Our theoretical analysis is on an annual timescale (Theorem 1). As stated in our citation (Schittekatte et al. 2024), "the per kWh rate is often constant for no less than a year and often much longer". Further, [price-per-kWh is indeed relatively constant (by less than a penny) annually](https://fred.stlouisfed.org/series/APU000072610).
> > >
> > > > **W5.3:** sounds like they actually have empirical results but it seems their results are merely simulations under some assumptions
> > >
> > > **(Response to W5.3):** Due to space, we refer you to "Empirical Analysis (Also Question 1)" in our rebuttal for Reviewer afca.
> > >
> > > ### Citations
> > >
> > > [C1] D'Arcangelo, Filippo Maria, and Marzio Galeotti. "Environmental policy and investment location: The risk of carbon leakage in the EU ETS." Energy Policy, 2025.

---

### Official Review · Reviewer_afca · 2026-03-12

**Significance:** 4
**Argument Clarity:** 3
**Rating:** 5
**Confidence:** 3

**Questions:**

1) Do you perform an empirical analysis, or does this merely refer to the plots of the deduced functions? If yes, this empirical analysis or simulation should be described and explained.

**Alternative Views Section:**

Yes

**Compliance With Llm Reviewing Policy A Conservative:**

Affirmed.

**Discussion Potential:**

4

**Final Justification:**

My initial evaluation was already positive, and the authors clarified the remaining issues and promised to revise the paper accordingly. Hence, I will keep my evaluation that this is, in my opinion, a clear accept (5).

**Paper Summary:**

This paper argues that AI developers and deployers should be incentivized to improve the efficiency of their AI systems through market-based mechanisms.
Current frontier AI models are usually trained using an unrestricted "hyperscaling" approach, which means that their quality largely depends on training with huge amounts of data. This leads to an increased need for large-scale GPU compute power.

The authors identify two main concerns with this approach: (1) Accessibility. The current trend creates an imbalance between large companies on the one hand, and academic institutions and small companies on the other. While the first group is able to afford the ever-growing need for compute power, the second group is increasingly left behind, which in turn further widens the gap between these two groups and leads to brain drain away from academia and toward big tech industry.
(2) Sustainability. The increasing demand for, and use of compute power entails serious environmental concerns.

To address these problems, the authors propose incentivizing more efficient AI development through a market-based approach. They present some known market-based methods and identify the cap-and-trade framework as suitable for their needs.
More specifically, they propose a cap on FLOP usage during AI inference only to avoid hindering research. The cap enables regulation of AI-related emissions, addressing the sustainability challenge. The possibility to trade surplus allowances especially supports small companies and startups, leading to healthier competition, hence, tackling the accessibility issue.
Allocation of caps is determined through benchmarking based on the overall governance goals, the AI operator's current consumption, efficiency, and previous behavior.

The proposed framework is undergirded by formulas capturing the participating agents' utilities and a theoretical deduction of Equilibria with and without governance through the cap-and-trade framework. The results show that theoretically, the application of the proposed framework leads to reduced FLOP usage.

The paper is completed by stating and discussing potential alternative views.

**Position:**

Yes

**Position In Title:**

Yes

**Related Work:**

4

**Strengths And Weaknesses:**

_Strengths_
This is an easy-to-follow, nicely written paper with a clear reasoning.
The authors provide an extensive overview of related literature, and arguments are mostly thoroughly supported by references.
It is nice that the authors support their position by theoretically modeling and analyzing their proposed framework.


_Weaknesses_
Overall, I have a positive opinion of this paper. However, there are some important points that need to be clarified before publication:
- In two places, it is implied that an empirical analysis is performed (first in line 348 (right side) "In Figures 1 and 2, we simulate the equilibrium" and second in the conclusion in lines 435/436 (right side) "We validate our framework empirically"). But as far as I can tell, this refers to simple plots of the established formulas. In my opinion, the words "simulate" and "empirically" are misleading here.
- It is not entirely clear how the formulas are deduced. For example, where does the term $1/x^k$ in line 330 come from, and why is it an appropriate choice for the model? Or in Figure 1, why are you considering $b=\sqrt{a}$ in the right image?
- It would be a nice addition to have a short section at the beginning, describing the contents of the paper, especially to mention that there is a theoretical analysis in the end. Currently, this is only hinted at through the word "provably" in the abstract.

_Minor remarks:_
- line 32: "of of"
- line 26 (right side): redefinition of ronnaFLOP
- line 75/76 (right side): "use misuse"
- line 118: "incentivizes" -> incentives?
- line 207/208: "such airline traffic" -> such as airline traffic
- line 235 (right side): one "are" too many
- line 298: "Below, detail" -> Below, we detail
- line 326/327: "AI AI"
- formula (3) does not make sense the way it is written: on the left side, we maximize over x, on the right side, x is a free variable
- I do not understand the sentence starting in line 432. What do you mean by "loosening"?

**Support:**

3

---

> ### Author Rebuttal · Authors · 2026-03-30
>
> Dear Reviewer afca,
>
> Thank you for your thorough and thoughtful review.
>
>
> Below, we address all questions and weaknesses. Before beginning, we want to mention that we have fixed all of the grammatical issues raised within the reviewer's minor remarks. Thank you for catching these!
>
> ## Weaknesses
>
> ### Empirical Analysis (Also Question 1)
>
> > **W1:** In two places, it is implied that an empirical analysis is performed (first in line 348 (right side) "In Figures 1 and 2, we simulate the equilibrium" and second in the conclusion in lines 435/436 (right side) "We validate our framework empirically"). But as far as I can tell, this refers to simple plots of the established formulas. In my opinion, the words "simulate" and "empirically" are misleading here.
>
> > **Q1:** Do you perform an empirical analysis, or does this merely refer to the plots of the deduced functions? If yes, this empirical analysis or simulation should be described and explained.
>
>
> **(W1 & Q1 Response):** We agree that our use of "simulate" and "empirically" are misleading, and have removed them from the paper. While in our attached code we verify that solving the optimization problems, via optimization solvers, in Equations 3 and 5 yield our theoretical solutions in Equations 4 and 6, Figures 1 and 2 indeed plot the theoretically derived solutions and are not simulations.
>
> * We have added a figure showcasing that our theoretical solutions are optimal (and deviating from them will result in reduced utility).
>
>
> ### Deduced Formula
>
> > **W2.1:** It is not entirely clear how the formulas are deduced. For example, where does the term $1/x^k$ in line 330 come from, and why is it an appropriate choice for the model?
>
> **(Response to W2.1):** We denote $1/x^k$, $k>0$ as the relationship between the number of FLOPs $x$ and the loss of an AI model. This relationship is backed by our cited work (Kaplan et al. 2020, Hoffmann et al. 2022), which states LLM "loss scales as a power-law with model size, dataset size, and the amount of compute used for training". We have added an explicit mention of this rationale within Section 6.
>
> > **W2.2:** Or in Figure 1, why are you considering $b = \sqrt{a}$ in the right image?
>
> **(Response to W2.2):** The setting $b=\sqrt{a}$ is a realistic situation where the cost of purchasing FLOPs from another company $b$ is more expensive than the cost of using FLOPs $a$ (since $a \in (0,1)$ in our figures, $b = \sqrt{a} >  a$). Concretely, once a company maxes out their FLOP usage they would likely pay a premium to another company in order to use more. This rationale has also been added within Section 6.
>
> * We note that the x-axis label in Figure 1b should read "Cost-per-FLOP (a)". This has been fixed.
>
> ### Theoretical
>
> > **W3:** It would be a nice addition to have a short section at the beginning, describing the contents of the paper, especially to mention that there is a theoretical analysis in the end. Currently, this is only hinted at through the word "provably" in the abstract.
>
> **(W3 Response):** We agree, and thank the reviewer for their feedback. We have added a section describing the contents of the paper, including mention of theoretical analysis.

---

> > ### Author Rebuttal · Reviewer_afca · 2026-04-02
> >
> > I thank the authors for their thorough reply. I think this is a good paper that offers a first step towards addressing the stated fundamental problems. After reading the other reviews, I think it constitutes a basis for further interesting discussions.

---

### Decision · Program_Chairs · 2026-04-30

**Decision:**

Accept (regular)

**Comment:**

This paper contends that market incentives are needed to improve AI efficiency, in order to improve accessibility and sustainability. The paper is well-written, and the position is argued convincingly. The paper also states some credible alternative views, including stifling innovation and national security interests.

The main weakness seems to be the call to action, which proposes a particular cap-and-trade framework. As the reviewers note, this proposal relies on a very simple economic model, seems difficult to implement in practice, and could be gamed in various ways. However, this is also a highly non-trivial problem, and I appreciate the authors’ effort to provide a concrete framework as a starting point for discussion.